# Hand Hygiene during the Early Neonatal Period: A Mixed-Methods Observational Study in Healthcare Facilities and Households in Rural Cambodia

**DOI:** 10.3390/ijerph18094416

**Published:** 2021-04-21

**Authors:** Yolisa Nalule, Helen Buxton, Alison Macintyre, Por Ir, Ponnary Pors, Channa Samol, Supheap Leang, Robert Dreibelbis

**Affiliations:** 1Department of Disease Control, London School of Hygiene and Tropical Medicine, London WC1E 7HT, UK; Robert.Dreibelbis@lshtm.ac.uk; 2Division of Psychiatry, University College London, London W1T 7BN, UK; helen.buxton.20@ucl.ac.uk; 3Policy and Programs Division, WaterAid Australia, Melbourne 3002, Australia; alison.macintyre@wateraid.org.au; 4National Institute of Public Health, Phnom Penh, Cambodia; ipor@niph.org.kh (P.I.); leangsupheap@yahoo.com (S.L.); 5WASH and Health Division, WaterAid Cambodia, Phnom Penh, Cambodia; porsponnary@gmail.com (P.P.); samolchanna@yahoo.com (C.S.)

**Keywords:** neonatal infection, hand hygiene, behaviour change, Cambodia, post-natal care, newborn care, formative research, intervention design, health facility, household

## Abstract

Background: Globally, infections are the third leading cause of neonatal mortality. Predominant risk factors for facility-born newborns are poor hygiene practices that span both facilities and home environments. Current improvement interventions focus on only one environment and target limited caregivers, primarily birth attendants and mothers. To inform the design of a hand hygiene behavioural change intervention in rural Cambodia, a formative mixed-methods observational study was conducted to investigate the context-specific behaviours and determinants of handwashing among healthcare workers, and maternal and non-maternal caregivers along the early newborn care continuum. Methods: Direct observations of hygiene practices of all individuals providing care to 46 newborns across eight facilities and the associated communities were completed and hand hygiene compliance was assessed. Semi-structured interactive interviews were subsequently conducted with 35 midwives and household members to explore the corresponding cognitive, emotional and environmental factors influencing the observed key hand hygiene behaviours. Results: Hand hygiene opportunities during newborn care were frequent in both settings (n = 1319) and predominantly performed by mothers, fathers and non-parental caregivers. Compliance with hand hygiene protocol across all caregivers, including midwives, was inadequate (0%). Practices were influenced by the lack of accessible physical infrastructure, time, increased workload, low infection risk perception, nurture-related motives, norms and inadequate knowledge. Conclusions: Our findings indicate that an effective intervention in this context should be multi-modal to address the different key behaviour determinants and target a wide range of caregivers.

## 1. Introduction

An estimated 73% of global newborn deaths occur during the first seven days following birth [1]. Infections are the third leading cause of neonatal mortality, and facility-born newborns in particular are susceptible to healthcare associated infections (HCAI) throughout their first week of life [2,3,4]. As part of the global action plan to end preventable neonatal deaths by 2030, the World Health Organization (WHO) recommends the integrated delivery of essential newborn care along the pregnancy to postnatal continuum of care [5]. The continuum of care for newborns spans pregnancy, childbirth and post-delivery [6]. Adequate hand hygiene through hygienic births and clean post-natal care is the cornerstone of the WHO recommended strategies to prevent infection-related newborn death [5,7]. However, despite the implementation of various hand hygiene promotion strategies in healthcare facilities (HCF) [8,9], hand hygiene compliance by both healthcare workers, parental and other caregivers during birth and post-natal care [10,11] remains low.

Understanding the hand hygiene practices and their determinants along the continuum of care in the early neonatal period (birth through the first seven days of life) is crucial for the design and delivery of comprehensive interventions to improve hand hygiene during this critical window. For facility-based births, the early neonatal period spans multiple environments as the mother–newborn pair transitions from the delivery room to the post-natal care (PNC) room, and discharge to the home environment [5,12,13]. Each of these environments constitutes different contexts, with different primary caregivers, and different environmental, social and psychological drivers of hand hygiene practices important for newborn health [11,14,15,16,17,18]. As hygiene behaviours are critical at both the HCF and home environment during this critical period, there is a need for innovative, contextualised behaviour-centred approaches that can effectively target both settings.

There are knowledge gaps along the continuum of care that undermine the development of comprehensive improvement strategies. The WHO recommendation for newborns to stay at least 24 h after birth within HCF prior to discharge to the home environment [12] makes the PNC room a critical point along the care continuum. However, hand hygiene behaviour in this environment is under-investigated, as most studies focus on the immediate childbirth period [14,19,20], the neonatal period in the home environment [21,22,23,24,25] or the intensive care environment for sick and small newborns [26,27,28]. Studies in resource-limited settings show that newborn care is often conducted by both healthcare workers and family members in the HCF [11,29,30], and by a range of household members in the home environment [11,22,31,32]; however, there is limited attention given to understanding and targeting non-maternal caregivers’ behaviours in improvement strategies.

Reducing neonatal mortality is a priority for the Royal Government of Cambodia (RGC). Between 2000 and 2014, maternal, child and infant mortality rates declined by over 80% in Cambodia, yet neonatal mortality rates declined by less than 50%, in part due to a persistent burden of neonatal infections [33,34]. Infections are the country’s third leading cause of neonatal mortality and accounted for 16% of all neonatal deaths in 2018 [35]. Over 80% of births in Cambodia are institutional deliveries [33] and the Cambodia’s Safe Motherhood Clinical Management Protocol for Health Centres recommends stays of at least 48 h following birth to ensure that the new mother and newborn receive adequate post-natal care in the facility [36]. Addressing infection risk factors in facilities and for facility-born neonates is therefore critical to reducing overall neonatal infection rates. Multiple studies have also highlighted gaps in water, sanitation and hygiene (WASH) conditions and behaviours and infection prevention and control (IPC) practices at the institutional and household level that place newborns at risk of infection [37,38,39,40].

The main objectives of this observational study were to understand the existing hand hygiene practices among healthcare workers, mothers and family members during the early post-natal care period in the HCF and home environment, and to explore the factors facilitating and hindering the observed practices. This mixed methods formative research study is part of the larger Changing Hygiene Around Maternal Priorities (CHAMP) project—a partnership among LSHTM, WaterAid Cambodia and the National Institute of Public Health, Cambodia to design and test an intervention targeting hand hygiene during childbirth and the early post-natal period in Kampong Chhnang Province, Cambodia. Findings from this formative study have been used to inform the design of the relevant components of the subsequent intervention. Findings related to hygiene during childbirth were reported in an earlier publication [41].

## 2. Materials and Methods

### 2.1. Study Setting and Sampling

This formative research study was conducted in Kampong Chhnang province between February 2019 and September 2019. Kampong Chhnang is located in central Cambodia and has a total population of approximately half a million people, [42] predominantly living in rural areas (~80%). The province is served by 42 primary health centres (PHC) and two referral hospitals (RH).

The study followed an explanatory sequential design where quantitative observations (direct and structured) informed subsequent in-depth qualitative data collection (semi-structured interviews) [43]. The overall study was guided by the Behaviour Centred Design (BCD) approach, which combines theory-based, ecological–evolutionary understanding of human behaviours with a systematic process for intervention development and evaluation [17,44]. Eight health facilities—6 PHC and 2 RH—were purposively selected for this observational study. As an exploratory study, the sample size was based primarily on time and resource constraints. To ensure a sufficient number of observations, the PHC that had the highest number of monthly deliveries were selected. The facilities were also selected to reflect different catchment areas across the province. Both referral hospitals located in Kampong Chhnang Province were included in the study.

Further details on the overall study context, site selection and sampling were described in an earlier publication [41].

### 2.2. Quantitative Methods

#### 2.2.1. Data Collection

Enrolment procedures and sample size for the observation data collection are described in Nalule et al. [41]. For a period of 14 days in the PHC, any eligible woman was invited to participate, up to a total of five consenting women per PHC. In the referral hospitals, there was no limit to how many women could be recruited over the 14-day observation period. Explicit mention of investigating handwashing behaviour as the aim of the study was avoided during recruitment of all participants to minimise reactivity.

Hand hygiene specific to newborn care was assessed through structured observations. Observations began when the mother–newborn pair was transferred to the PNC room of participating facilities. Over a period of four continuous hours, data collectors chronologically recorded newborn care practices (diaper changes, cord care, breastfeeding and general newborn handling), and any corresponding hand hygiene and gloving practices of all caregivers, defined as any individual providing newborn care during the observation period. Following the end of each post-natal observation, consenting women provided their contact details and home address and agreed to a convenient time and day for a home visit within 72 h after observation at HCF. Home observations were conducted only for the women recruited in the six PHC and lasted six continuous hours from the time of the data collector’s arrival at the home. Similarly to the PNC, home observations included newborn care practices and hand hygiene practices of all caregivers.

In addition to the direct observations, facility-level and household structured assessments were conducted to assess water, sanitation and hygiene (WASH) conditions of the HCF and the household. The facility-level tools have been described in an earlier publication [41]. Structured household assessments were conducted by the data collectors either upon arrival or at the end of the home observations.

The data collectors were made up of six female healthcare workers, divided into two teams. Prior to data collection, all observation tools were piloted and iteratively refined during a one-week training period in two HCF in the same province which were not part of the study sample. Refresher training was conducted prior to the data collection at the referral hospital. Each team was assigned to a single facility and completed observations over three different shifts, covering a full 24-h period.

#### 2.2.2. Data Analysis

All observation data were analysed using StataSE 15 (Stata Corp, College Station, TX, USA). Any qualitative text entries were reviewed, and where applicable, recoded quantitatively, and analysed as structured observation data. Data analysis was descriptive and focused on the frequency and sequence of hand hygiene opportunities and associated hand hygiene actions. Newborn care-related hand hygiene opportunities were based on WHO’s Five Moments for Hand hygiene [45,46], WHO postnatal care recommendations [12,13] and the three moments adapted for neonatal hand hygiene in the community described by Ditai et al. [23]. Hand hygiene opportunities included prior to newborn contact, prior to clean/aseptic procedure (cord care, injection/immunisation) and after cleaning the newborn’s bottom following defecation.

For each caregiver, hand hygiene actions associated with each hand hygiene opportunity were coded into two categories aligned with the available literature and WHO guidelines [12,13,23,45,46] for the analysis: adequate (handwashing with soap and water and/or use of alcohol-based hand rub; handwashing with soap and water plus glove use for any aseptic/clean procedures) and inadequate (handwashing with water only; wearing gloves without intermediate handwashing with soap; or no observed hand hygiene action taken). A caregiver’s hand hygiene’s category could vary throughout the course of the observation, depending on hand hygiene opportunities observed and actions taken at a particular point in time.

In our analysis, caregivers of the newborns were categorised into four groups; mothers, fathers, healthcare workers (midwives, nurses, doctors and interns) or non-parental caregivers (all other individuals observed providing care to the newborn). Newborn care was further categorised into two groups for analysis: observed cord contact and other newborn contact. “Other newborn contact” included any other hand hygiene opportunity where the individual made contact with the newborn outside observed contact with the umbilical cord.

Descriptive statistics were used to analyse the frequency and proportion of hand hygiene opportunities that appeared/were utilised under each hand hygiene category: by caregiver and type of newborn care. Data from the home and facility level assessments were analysed descriptively and triangulated to provide context to the structured observations and insights into the subsequent qualitative data findings.

### 2.3. Qualitative Methods

#### 2.3.1. Data Collection

Findings from the quantitative observations were reviewed by project stakeholders during a 2-day framing workshop (22–23 August 2019, Phnom Penh). The key behaviours of interest identified for in-depth qualitative investigation were hand hygiene around newborn care in the PNC room and the home environment. The identified key targets for behavioural change were midwives, parents and non-parental caregivers.

The data collection took place over a period of 2 weeks in September 2019. Semi-structured interviews were conducted with midwives at the HCF and with parental and non-parental caregivers in their home environment. The sample size was based on the anticipated number required to reach theoretical saturation. Semi-structured interview (SSI) tools were designed to further investigate barriers and opportunities for target behaviour uptake and inform a gender analysis of individual and household domains. Within a single SSI, additional formative research tools were completed [17] to actively engage the participants. Details of the specific tools are described by Nalule et al. [41]. The formative research tools used depended on the specific respondent and the time available for the interviews.

Mothers were recruited for the qualitative component of the study as they waited for discharge in the PNC room. Eligible participants were women who had a vaginal birth at the HCF, with no maternal or newborn complications, were waiting for discharge and lived within one-hour travel time of the HCF. Consenting mothers were interviewed at their houses approximately one week after discharge. At the household, the data collector could recruit up to 2 more participants (father and a non-parental caregiver) for additional individual interviews.

All interviews were conducted in Khmer language by two teams of two female Cambodian enumerators who had prior experience in qualitative data collection. Qualitative tools were tested and refined over a three-day training period. All interviews were audio recorded, and free form notes and pictures of completed activities were taken. Summaries were recorded in a semi-structured data capture form, following data collection and daily debriefing sessions.

#### 2.3.2. Data Analysis

All qualitative data analysis was done using Microsoft Word and Excel (Redmond, Washington). Initial analysis of the preliminary data (all field notes, written response summaries and any salient findings from daily debriefs) was entered into a spreadsheet and organised by data collection tool and activity. Study team members verified data entry, and audio recordings were consulted for clarity or further exploration. All data (spreadsheet and audio recordings) were coded, organised and analysed against pre-defined categories of behavioural determinants based on the Behaviour Centred Design (BCD) checklist adapted for handwashing behaviour [47]. See Appendix A for more details.

## 3. Results

### 3.1. Maternity Setting Hand Hygiene Conditions

None of the maternity settings in the eight HCF had functional hand hygiene facilities (with water and soap and/or alcohol-based hand rub) at all points of care (delivery room, post-natal care room, waiting area and toilets) (Table 1).

Handwashing stations were located at the toilets of all eight maternity settings and 75% (6/8) had soap present at the time of observation. All eight delivery rooms had functional handwashing facilities, all of which were restricted for staff members’ use. Half (4/8) of the HCF had a handwashing station for use by mothers, fathers and non-parental caregivers within the maternity setting, but only one had soap available. Only one facility had a handwashing station located inside the PNC room. No hand hygiene alternatives (e.g., alcohol-based hand rub (ABHR)), hand drying materials or hand hygiene posters were available for mothers, fathers and non-parental caregivers in the eight HCF.

### 3.2. Structured Observations (PNC Room)

#### 3.2.1. Participant Characteristics

A total of 46 mothers and newborns were enrolled in PNC room observations—22 from the primary health centres and 24 from the referral hospital (Table 2).

Observations in the PNC room occurred an average of 1.4 h after birth (range: 1–5); 45 of these mothers had been previously observed during labour and delivery, and one additional mother was recruited while in the PNC room at the referral hospital. Mothers had similar characteristics across the observations, with a mean age of 28 (range: 21–40) and had an average of 2 (range: 0–6) previous live births.

#### 3.2.2. Hygiene Opportunities and Actions

On average, one healthcare worker (range: 0–4), usually a midwife, provided care to the mother and newborn in the PNC room during observation periods (Table 2). In 7% (3/46) of observations, no healthcare worker visited the mother–newborn pair during the observation period. Mothers and newborns were visited by an average of four people (range: 1–8). Across the observations, fathers were the most common non-maternal caregiver, present in 82% (38/46) of observations. Among the non-parental caregivers, the grandmother was most commonly present (32/46).

Newborn care activities resulted in a total of 811 hand hygiene opportunities observed in the PNC room (Table 3).

Non-parental caregivers accounted for over half (57%) of hand hygiene opportunities. The remaining hand hygiene opportunities during newborn care were among mothers (31%), fathers (9%) and healthcare workers (3%). When disaggregated by type of newborn care, non-parental caregivers were responsible for 100% of the hand hygiene opportunities related to direct cord contact events.

In none of the newborn care activities was hand hygiene practiced adequately across all the caregiver groups.

### 3.3. Structured Observations (Home)

#### 3.3.1. Home Characteristics

Of the 22 households observed, 17 had a designated place for handwashing within the compound, and 11 of households had soap present at the handwashing site (Table 4).

Of the 17 observed handwashing facilities, 13 (77%) were buckets to pour over hands, two (12%) were sinks with taps and two (12%) were bowls for handwashing. The handwashing facilities were located an average of five metres (range 5–20m) away from the mother–newborn pair.

#### 3.3.2. Participant Characteristics

Only mothers observed in the PHC (*n* = 22) were observed at home. All mothers consented to the home observations and there was no loss to follow up. Over half (55%) of the home observations took place on the day of discharge from the facility, and the remainder (45%) one day after discharge (Table 2).

#### 3.3.3. Hygiene Opportunities and Actions

Over the six-hour observation period, there were on average seven people (range: 2–13) present with the mother–newborn pair (Table 2), many who participated in at least one newborn care activity. Similarly to the PNC observations, the father (20/22) was the most common non-maternal caregiver present during the home observations.

Of the 508 hand hygiene opportunities related to newborn observed during the observation period, mothers (50%) and non-parental caregivers (46%) accounted for the majority of these opportunities (Table 5).

Similarly to the observations in the PNC room, most of the direct cord contact activities (92%) were conducted by non-parental caregivers. No hand hygiene practice was conducted by any newborn caregiver at any opportunity.

### 3.4. Qualitative Results

#### 3.4.1. Participant Information

Semi-structured interviews were conducted with 13 midwives across four of the observed HCF (three primary health centres and one referral hospital). Two households were interviewed per HCF for a total of eight households. Within each household, three household members—a mother, father and grandmother—were interviewed. Two grandmothers did not consent and were not interviewed. In total, eight mothers and fathers and six grandmothers were interviewed.

Interviews provided insights into the behavioural determinants that influence hand hygiene practices in the PNC room and the home. The findings are summarised against the key BCD components. See Appendix A for more details.

#### 3.4.2. Behaviour Setting for Newborn Care

**Stage:** In the PNC room, newborn care typically took place around the mother’s bed where the newborn was always located. The mother spent most of her time at the facility lying in bed, getting up primarily to breastfeed and to attend to her own personal hygiene. During the first week in the home environment following discharge from the HCF, the mother–newborn pair spent most of their time either outside on the veranda of the house or in the bed that was located on the ground floor of the house. The majority of the newborn care also took place at that location.

**Roles and Norms:** Midwives reported their main responsibility in the PNC room to be monitoring the mother–newborn pair; supervising cleaning staff; and when necessary, supervising the father and non-parental caregivers. Midwives regarded their position as the most authoritative and highly respected in the maternity ward and therefore did not find it difficult or uncomfortable to immediately correct all other caregivers’ behaviours following observed non-compliance to hygiene practices.

Fathers and non-parental caregivers providing direct care for the mother–newborn pair during their stay in the PNC room was the norm. Fathers and non-parental caregivers, primarily the grandmother, assisted with the majority of the maternal and newborn-caregiving activities, typically staying with the mother–newborn pair at the bedside until discharge. Mothers, fathers and non-parental caregivers reported trusting the midwives’ advice and guidance around the best newborn care practices over any contradicting advice given by other family members. Some grandmothers reported that they would follow a midwife’s hand hygiene advice even when they did not agree with it. All household members expressed discomfort and unwillingness to correct the midwives in scenarios where inadequate hygiene practices by the midwives was observed.

Following discharge, mothers reported taking a more active role in newborn care activities. In line with the direct observations, the grandmother and father reported that they were also involved in newborn care at the home, including cord care, diaper changing and other household tasks whenever the mother was unable. Mothers expressed difficulty enforcing adequate hand hygiene practices with non-parental caregivers due to existing family hierarchies and dynamics. However, mothers reported willingness to risk upsetting this dynamic and correct family members if the advice was given to them by the midwives. Unlike the new mothers, grandmothers and fathers reported not experiencing any difficulty or discomfort in correcting any observed noncompliant behaviour by other non-parental caregivers in the home environment or in the PNC room.

The participation of fathers in newborn care activities reflects a temporary shift in normative behaviour around housework. Household tasks in our setting were typically divided according to the prevalent descriptive gender norms. All respondents described childcare responsibilities and daily household tasks such as cooking, house cleaning and laundry as activities only women did in the household. The early neonatal period was highlighted as the exceptional circumstance when men would take on “women-only” roles and participate in newborn care activities, including diaper changing, cord care, bottle feeding and newborn bathing. Other exceptional circumstances included when the woman was ill or away from home for an extended period of time. This prevalent normative practice of gendered division of household tasks, however, did not align with the respondents’ personal beliefs. All household respondents described the expectation of the man participating equally in all household activities on a regular basis.

**Props and infrastructure:** The lack of accessible handwashing infrastructure and related materials was the most common barrier faced by mothers and other caregivers to practicing good hygiene at the health facility. All household respondents found handwashing facilities at HCF inaccessible and inconvenient relative to the location of newborn care. The locations of handwashing facilities, including those that were located outside by the PNC entrance, were particularly challenging for mothers and caregivers providing continuous bedside care. Mothers reported mobility difficulty due to pain, discomfort and fatigue following recent delivery, limiting their ability to access distant handwashing facilities. Other handwashing challenges reported by household respondents included the unavailability of soap at sinks, the lack of running water due to broken taps and crowded conditions within the PNC room.

#### 3.4.3. Brain and Body Factors

**Knowledge:** Knowledge around adequate hand hygiene differed between midwives and the household respondents. Midwives viewed hand hygiene as requiring both soap and water to be effective. In contrast, most mothers, fathers and non-parental caregivers considered hand rinsing (using water only) as sufficient hand hygiene practice for newborn care. Both the midwives and household respondents stressed the importance of hand hygiene prior to cord contact and prior to breastfeeding, but handwashing prior to newborn contact or following diaper changes was rarely mentioned.

Midwives reported relaying hand hygiene information to mothers and other caregivers during two moments in post-natal care: shortly after birth and during facility discharge. Midwives did not actively communicate hand hygiene information during a family’s stay in the PNC room, reporting instead that hand hygiene promotion materials were sufficient for non-parental caregivers during that time. However, no hand hygiene promotion materials were observed in PNC rooms (Table 1), and mothers, fathers and other non-parental caregivers did not recall seeing any educational hand hygiene materials at the facility. Mothers reported being given advice about hand hygiene at the time of discharge; however, fathers and other non-parental caregivers were either not present for discharge instructions or reported not paying attention. The most recalled hand hygiene information by the mother was around cord care, breast cleaning prior to feeding and perineum care.

**Risk perception**: All respondents perceived the risk of infection in the PNC room to be very low. Activities such as entry to the PNC room from outside the HCF or the delivery room and newborn handling were considered low risk to the newborn, particularly when the baby was wrapped. Umbilical cord care was the only newborn activity that was considered by both midwives and household respondents as a high-risk caregiving activity that could result in infection to the newborn.

**Discounts:** Midwives reported having no time for frequent follow-up discussions with mothers and other caregivers, and limited time to supervise hygiene practices in the PNC area. This lack of time was most acute when staff were limited, particularly at night. More urgent labour and delivery activities, providing antenatal services and doing administrative work, were prioritised over PNC room oversight responsibilities. Midwives would only prioritise roles related to the PNC room during post-birth complications, discharge periods and at the end or beginning of their shifts.

The lack of time available for hand hygiene and prioritising other activities extended to the household setting. Mothers, fathers and non-parental caregivers reported being too busy to wash their hands at all recommended times due to increased workloads and responsibilities around childcare in both the PNC and at home. In addition to caring for the newborn, the respondents reported simultaneously performing other caregiving or household-related tasks, such as caring for the new mother, cleaning the PNC room and home, food preparation, laundry activities and caring for other children.

**Motives:** Nurture was both a facilitator and a barrier to practicing good hand hygiene among household respondents. All household respondents commonly cited having a happy and healthy baby (nurture) as their reason to practice good hygiene. Conversely, household respondents commonly pointed to nurture-associated emotions superseding their practice of proper hand hygiene practices. Feeling worried or concerned, caregivers prioritised immediately alleviating the newborn’s perceived distress and would either skip or forget handwashing steps in order to quickly pacify the crying newborn. Fathers and non-parental caregivers both in the PNC room and home would forget to wash their hands in a rush to make contact with the newborn because of the joy and excitement at seeing their newborn relative.

**Senses:** Household members typically relied on visible contamination to cue hand washing behaviours during newborn care. The absence of dirt on the hands signified cleanliness and household members did not feel the need to wash visibly clean hands before holding the newborn, even when coming from outside.

## 4. Discussion

This study was designed to explore hand hygiene practices and the behavioural determinants around early newborn care in the PNC room and at the home in rural Cambodia. Our findings show a high frequency of hand hygiene opportunities by a wide range of caregivers with minimal hand hygiene compliance. Among all the groups involved in caregiving, non-parental caregivers accounted for the majority of hand hygiene opportunities. The hand hygiene practices of midwives and other caregivers during newborn care in the PNC room and home were influenced by a range of factors, including a lack of physical infrastructure and supplies for handwashing, inadequate knowledge, low infection risk perception, nurture-based motivations, norms, the absence of hand hygiene reminders, limited time and a high workload. Existing studies assessing hygiene practices in Cambodia during newborn care in healthcare facilities and at home are limited and utilise self-reporting and proxy measures of handwashing behaviour [37,38]. Our study adds to this limited literature and strengthens the available evidence by using detailed direct observations, the recommended gold standard, to quantify these practices [48,49].

The lack of physical opportunities was a significant hand hygiene barrier faced by new mothers, fathers and non-parental caregivers in the PNC room. Facility-based studies in Cambodia have similarly highlighted the lack of functioning and accessible handwashing facilities as a major barrier to improved hand hygiene in health facility settings [40], and more specifically during newborn care in the maternity units following birth [37,38]. Formative research findings of ten HCF in Cambodia by Bazzano et al. [38] found post-partum women and their families had no access to handwashing stations for newborn care in 90% of the surveyed facilities. The consistent provision of alcohol-based hand rubs (ABHR) is a convenient, economical and effective alternative strategy that could be employed to improve hand hygiene compliance for new mothers and other caregivers [50,51,52]. Facility-based studies looking at the relationship between increased physical opportunities and use among visitors and patients in low resource settings are limited. However, studies in high-income settings have found ABHR to be associated with improved hand hygiene when conspicuously placed at point of care areas such as at the bed and at the entrance to the room, and when using mobile dispensers. [50,53,54,55]. Strategic placements of ABHR in our settings would not only increase the convenience of hand hygiene practice for the mobility restricted mother but also would address handwashing barriers faced by the paternal and non-parental caregivers, such as restricted movement due to overcrowding in the PNC room and time pressure from the urgent caretaking needs and increased workload.

Combining the availability and accessibility of physical opportunity with strategically placed cues in the environment to trigger timely handwashing has a more sustained effect on hand hygiene compliance in institutional settings [56,57,58,59,60]. Attention-grabbing visual [61] or auditory cues [62] placed around the handwashing facilities and the location of the mother–newborn pair in our setting could serve as both guidance and reminders of how and when to correctly practice hand hygiene during newborn care. Nurture was a strong motive for handwashing behaviour in our study and could be utilised in the development of these reminders as well as broader messaging and educational strategies. The nurturing emotions have previously been identified as an important driver of caretakers’ handwashing behaviour [31,63,64] and have been leveraged as part of effective hygiene interventions in India [65] and Nepal [66]. Designing cues to evoke and associate positive nurturing emotions with hand hygiene practice could motivate and further enhance the practice of the desired behaviours [65,67].

Paternal and non-parental caregivers played significant roles in early newborn care both in the HCF and at home in our study. The frequent involvement of a wide range of caregivers in patient care activities is consistent with other facility-based studies in countries in Asia and Africa [11,29,30,59,68]. Influenced by social, religious, cultural and institutional factors, caregivers spend long periods of time in the HCF engaging in invasive and non-invasive patient contact and often with inadequate hand hygiene [11,29,30,59,68]. Family members providing in-patient care in a hospital in Bangladesh accounted for 54% of all hand hygiene opportunities with a 2% hand hygiene compliance rate [59]. In Nigeria, during newborn care in PNC rooms across three HCF, non-maternal caregivers accounted for 64% of all hand hygiene opportunities with a 0% hand hygiene compliance rate [11]. There is a limited understanding of the role of paternal and non-parental caregivers’ hands in the carriage and transmission of HCAI to neonates [29,30,59], and these groups continue to be overlooked by IPC guidelines and strategies. A review by Park et al. [29] found that despite family caregiving being the norm in South Korea, Indonesia and Bangladesh, only six out of 92 HCAI policies and guidelines across the three countries acknowledged the role of family caregivers, and only one guideline recommended their inclusion in the IPC strategies.

In addition to their engagement in newborn care, fathers and non-parental caregivers in our study also played a key role in influencing the handwashing behaviours of other caregivers, depending on their position along the family’s hierarchical structure. Similar handwashing studies in Bangladesh and Indonesia have highlighted the role of existing family hierarchies and social norms in supporting [22] or hindering [31] handwashing behaviours in the home environment, particularly among new mothers. In Bangladesh, Parveen et al. [31] found that new mothers’ handwashing practices were hindered primarily by a lack of support of the more influential family members to change existing handwashing norms or create environments enabling physical handwashing. In Cambodia, mothers in the domestic sphere typically occupy limited influential roles within their own families and households [69] and may not be in the position, however willing, to motivate handwashing behavioural changes among other family members. In addition to targeting caregivers to practice the recommended hand hygiene practices, a more effective intervention would further engage specific family members such as the father and grandmother as authorities to inform and influence these hygienic practices among other relatives and visitors.

The level of inadequate hand hygiene at the household indicates the continued potential risk of pathogen transmission to newborns following facility discharge. Hand hygiene improvement strategies need to ensure that the behaviours introduced in the facility are maintained following the transition to the home environment. Intervention studies targeting household behavioural changes are typically designed with some or all of the intervention components delivered at the community level [21,65,66,70,71]. Facility-level interventions, however, have shown promising results as an alternative approach to improving and sustaining WASH related behaviour and health outcomes at the household level [72,73,74,75]. Interventions at HCF target household members at a critical time when they are much more receptive and motivated to change behaviours due to experiencing a heightened perception of health risk and the perceived benefits of preventive health behaviours [76]. One’s first pregnancy and early parenthood is a similar period of increased receptivity to changing behaviours due to increased risk perception, changes in self-definition and societal position and increased nurture-based responses [22,31,76,77], and has been successfully utilised in facility-level interventions to improve WASH behaviours in households of both participants and non-participant neighbours and friends [73,74,75]. Integrating intervention delivery along the healthcare continuum provides multiple touchpoints for contact with a wide range of caregivers and opportunities to repeat and reinforce messaging strategies without creating additional responsibilities for low staffed HCW with high workloads.

Our study found that despite men’s increased participation in caregiving and household tasks in the period following childbirth, the hand hygiene information given at the HCF maintained the prevalent gender norms, targeting primarily the mother as the assumed sole primary caregiver and overlooking the involvement and engagement of the father. Engaging all caregivers provides an opportunity to challenge existing gender stereotypes of women as sole caregivers with responsibilities for household duties and childcare. Behaviour change communications and promotion in our setting should be responsive to the shifting gendered responsibilities that occur during this time and intentionally include fathers alongside other identified family members as primary newborn caregivers for more effective outcomes [78]. Midwives were the most respected and trusted source of health information and had the most frequent contact with mothers and other caregivers, and should be trained as the key healthcare workers for these behavioural change communications.

This study had several limitations. The small number of facilities for this observational study limits the generalisability of our findings to beyond these study sites. The use of the BCD theoretical framework to inform the formative research, however, allows for generalisable lessons to be taken from context-specific settings. Our observation periods were limited in duration, and despite observing several hundred hand hygiene opportunities, we only observed a limited number of very high-risk events, such as cord contact and cord cleaning. Our home observations were limited to mothers who attended the primary health centres, and while our data suggest that these mothers had similar characteristics, there may be some unexplored systematic differences between mothers in referral hospitals and primary health centres that may have introduced bias into our study. Participant reactivity during the observations may have led to an overestimation of hand hygiene compliance. We attempted to minimise this by avoiding any explicit mention of measuring hand hygiene compliance during enrolment and carrying out the qualitative interviews after all the structured observations were completed. The complete lack of observed hand hygiene during the observation period suggests that any reactivity on the part of study participants was minimal.

## 5. Conclusions

Our formative study provides a comprehensive picture of hand hygiene practices and the potential infection risk faced by the newborn during the early neonatal period. Combined with our previous findings of low hand hygiene compliance during newborn care in the delivery room immediately after birth [41], newborns are at high risk for infection, and multi-component interventions along the entire continuum of care are essential to address hand hygiene practices and the key determinants of a wide range of caregivers. Our findings indicate that a multi-modal hand hygiene intervention delivered at the facility that creates an enabling physical and social environment to facilitate the performance of the desired behaviour and incorporates cues (environmental and verbal) to guide, remind and reinforce practice during this teachable moment for a wide range of caregivers, could improve hand hygiene behaviours in both the HCF and the home.

## Figures and Tables

**Table 1 ijerph-18-04416-t001:** Maternity setting hygiene conditions.

	Primary Health Centre (*N* = 6)	Referral Hospital (*N* = 2)
**Functional hand hygiene facilities at all points of care in maternity setting**	0	0
**Handwashing facilities within maternity setting (not staff-restricted)**		
Available	2	2
Soap and water/ABHR available	0	1
**Handwashing station at toilet**		
Available	6	2
Soap and water/Alcohol based rub available	4	2
**Handwashing stations in delivery room**		
Available	6	2
Soap and water/Alcohol based rub available	6	2
**Handwashing facilities inside PNC room**		
Available	0	1
Soap and water/Alcohol based rub available	0	0

**Table 2 ijerph-18-04416-t002:** Participant characteristics.

Facility Type	Primary Health Centre (*n* = 22)	Referral Hospital (*n* = 24)
**Post-natal care room**	**Mean (range)**	**Mean (range)**
Age	28 (21–40)	28 (21–38)
Previous live births	1.4 (0–6)	1.6 (0–6)
Time travelled to HCF (min)	17 (5–40)	22 (5–40)
Time elapsed since birth (hours)	1.4 (1–3)	1.5 (1–5)
Number of PNC visitors	4 (1–8)	4 (2–7)
Number of healthcare workers	1 (0–4)	1 (0–3)
**Home**	**Mean (range)**	**Mean (range)**
Days elapsed since birth	2 (1–3)	n/a
Days spent at home since discharge	0 (0–1)	
Number of visitors present	7 (2–13)	

**Table 3 ijerph-18-04416-t003:** Observed hand hygiene opportunities and hand hygiene actions in the post-natal care room.

	Hand Hygiene Opportunities (*N*)	Hand Hygiene Category *N* (%)	
		Adequate ^1^	Inadequate ^2^
**All newborn care**			
Healthcare workers	22 (3%)	0 (0%)	22 (100%)
Mothers	251 (31%)	0 (0%)	251 (100%)
Fathers	73 (9%)	0 (0%)	73 (100%)
Non-parental caregivers	464 (57%)	0 (0%)	465 (100%)
**Total**	**811**	**0 (0%)**	**811 (100%)**
**Newborn care (cord contact) ^3^**			
Non-parental caregivers	3 (100%)	0 (0%)	3 (100%)
**Total**	**3**	**0 (0%)**	**3 (100%)**

^1^ The adequate hand hygiene category includes washing hands with soap/washing hands with soap and wearing gloves for aseptic procedures. ^2^ The inadequate hand hygiene category includes no hand hygiene action, rinsing hands without using soap and wearing gloves without handwashing with soap prior to donning gloves. ^3^ Cord contact includes direct cord contact via cord cleaning or cord inspection.

**Table 4 ijerph-18-04416-t004:** Home WASH conditions.

	*N* (*N* = 22)	%
**Hand hygiene items available in household**		
Soap	21	95.5
Detergent	21	95.5
**Handwashing facility**		
Available	17	77.3
Soap/Detergent available at site	11	64.7
Has hand drying materials	1	5.88
**Handwashing type**		
Sink with tap	2	11.8
Bucket to pour over hands	13	76.5
Bowl to wash hands in	2	11.8

**Table 5 ijerph-18-04416-t005:** Observed hand hygiene opportunities and hand hygiene actions in the household.

	Hand Hygiene Opportunities	Hand Hygiene Category *N* (%)	
	*N*	Adequate	Inadequate
**All newborn care**			
Mothers	246	0 (0%)	246 (100%)
Fathers	35	0 (0%)	35 (100%)
Non parental care-givers	227	0 (0%)	227 (100%)
**Total**	**508**	**0 (0%)**	**508 (100%)**
**Newborn care (cord contact)**			
Mothers	1	0 (0%)	1 (100%)
Non parental caregivers	11	0 (0%)	11 (100%)
**Total**	**12**	**0 (0%)**	**12 (100%)**

## Data Availability

The datasets used and/or analysed during this study are available from the corresponding author on reasonable request.

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
