# Peer review of "Hand Hygiene during the Early Neonatal Period: A Mixed-Methods Observational Study in Healthcare Facilities and Households in Rural Cambodia"

_ijerph, 2021, doi:10.3390/ijerph18094416_

Round 1

Reviewer 1 Report

This study investigates the behaviours of caregivers and family members with relation to hand hygiene in the early neonatal period. The manuscript is well written in general.

The introduction is a little bit long and can be reduced.

I have some comments regarding the methods section:

-Please explain the overall study context, aims and study site selection and sampling even though they were described in a previous publication.

-Was the caregivers/ family members blinded during the first phase of the study (the quantitative phase)?

-Who were the people who did the interviews? Do they speak the local language or was there a translator to help?

-When interviews were done with more than a family member in the same household, were the interviews done separately or with all the family members at once?

Reviewer 2 Report

Overall

This is a very thoughtful study whose conclusions are that there is a need for:

  • Improving washing facilities
  • Better knowledge among carers

All of these findings are very important and add knowledge to the field. However, this study and other similar studies show that there is a need for an interventional study to tackle the clinical problem.

I recommend the authors are clear about the study approach:

  1. It is an observational study
  2. Main methods are structured observations by the research team and interviews with participants
  3. The research aim is to provide a better understanding of handwashing provisions and practise among care givers.

Please provide a clear research question early in the manuscript.

Also, I recommend the authors highlight what is novel about this study.

Introduction

Well written and clear background about the importance of the hand hygiene.

Can the authors consider the difficulties in educating staff and families and implementing change in the context of the primary and tertiary care settings? What interventional studies have been conducted before? Those referenced are mostly observational.

The end of the introduction explains that the study is a ‘mixed-methods, theory-driven formative research study… whose purpose is to investigate hand hygiene practices and determinants among healthcare workers, mothers and family members along the new-born’s entire care continuum’. Can the authors clarify for the average reader, if this is an observational study? What groups are being compared, if any?

What was the overarching research question?

Methods

Key information about study aims and context is referenced but it would be useful for the readers to have at least a summary of this information in the manuscript.

The description of how carers are observed is good. Other data collected including WASH is also appropriate. Optimising of observations and data collection prior to starting the study is also good.

For the categories of ‘adequate’ and ‘inadequate’ hand washing, is this based on clear evidence that these specific practises are associated with a reduced risk of neonatal infection? What are these definitions based on? This is important for the author’s eventual conclusions.

Results

The data suggests the venues (healthcare and homes) need to improve their provisions for hand washing. The majority of homes had reasonable WASH facilities. However, none of the care givers adequately washed their hands based on the author’s predefined categories.

The qualitative data is the most striking output from this study. There are clear misconceptions regarding hand hygiene. There is a good opportunity for an interventional study to improve these aspects of knowledge and investigate the effect on neonatal infections.

Discussion

Well written. The first paragraph is a very helpful and clear summary. Something similar in the introduction/methods is needed. However, this early paragraph is also an opportunity to explain what this study adds to the already existing observational studies published?

Has the research team considered their next project? How can the findings of this study be used to reduce the incidence of neonatal infections?
